# Being a military child in Denmark: Young people's experiences of living with a parent with PTSD

**Paul Watson**[1]*, **Alison K. Osborne**[2]

**1** Northern Hub for Veterans and Military Families Research, Faculty of Health and Life Sciences, Northumbria University, Newcastle upon Tyne, United Kingdom, **2** Faculty of Psychology, Northumbria University, Newcastle upon Tyne, United Kingdom

* paul5.watson@northumbria.ac.uk

**Data Availability Statement:** "Samples of the transcripts are provided in the paper and Supporting Information files. Given the sensitive nature of content and the detailed narrative

## Abstract

This article describes the impact parental combat related Post Traumatic Stress Disorder (PTSD) has on children and young people from military families, their relationships within their family dyad, their emotional health and well-being and their social connectedness. Moreover, this article, aims to understand the lived experiences of children and young people from military families to understand the impact the emotional health and well-being of military connected children and young people during a five-day residential camp in Denmark ran by Støt Soldater & Pårørende (SSOP) a charity who provide psychosocial intervention to support the emotional well-being, decision-making, confidence, resilience and self-esteem of military children and young people.

## Introduction

Denmark has deployed soldiers to the Balkans (1992–1999), Kosovo (1999-ongoing), Afghanistan (2022–2021) and Iraq (2003–2011). In contrast to some NATO countries, Denmark's first formal veteran policy enables support services for veterans who have been deployed on at least one internation operation. The person may continue to be employed in the Danish Defence, and Danish veterans might still serve on active duty, but may also have left the services [1]. Veterans in Denmark have access to the country's tax-funded welfare system, just like any other citizen. This includes services such as healthcare, hospital care, social services, and financial assistance. While the support system generally depends on veterans and their families to seek out available resources, most municipalities have a designated veteran coordinator. Their role is to assist veterans and their families in connecting with the appropriate support services within the welfare system [2, 3].

Depending on the social and political landscape of the country, children and young people from military families are often portrayed as a stereotyped group, rather than a multi-layered complex and heterogeneous population [4]. Military children and young people are exposed to a range of stressors that are rarely experienced by their civilian counterparts [5], such as, repeated and often extended periods of separation from a parent serving in the military.

structure of each interview, whole transcripts cannot be completely de-identified to be shared publicly. This is to protect the participants and their families in the study, as well as the national security of the Danish military. Queries regarding the data may be directed to Northumbria University Director of Ethics (Nick.Neave@northumbria.ac.uk) or Records and Information Manager (Ducan. James@northumbria.ac.uk)".

**Funding:** This study was funded by Støt Soldater & Pårørende (SSOP) and payed for both the authors PW and AO. The funder did not contribute to the data analysis or writing of this article.

**Competing interests:** The authors have declared that no competing interests exist.

Decent conflicts military personnel, their families and veterans face different physical and psychological issues. This includes surviving serious injuries due to improved weaponry, combat safety equipment and advances in medical treatment, alongside enhanced rehabilitation, military personnel are surviving serious injuries. However, whilst this enhancement on treatment and after care has significantly reduced combat fatalities, there is a rise in the number of amputees and those effected by blast and psychological injuries [6–9].

Since the war in Vietnam, military research has paid significant attention to Post Traumatic Stress Disorder (PTSD). Post Traumatic Stress Disorder (PTSD) is a diagnosis under the DSM-5 by cluster of symptoms including the following presentations: intrusion symptoms, avoidance, negative alteration in cognitions and mood, and alterations in arousal reaction [10]. Tanielian et al. [11] concluded that military personnel who served in Iraq had a marked increase in the presentation of depression, anxiety, and symptoms of PTSD, between three- and twelve-months post deployment. The stressors military personnel may face in combat has the potential to disrupt the cohesiveness of, and integration back, into their family. These stressors include, anger, excessive drinking and a want to return to the combat zone [12].

The prevalence of PTSD symptoms often increases overtime with delayed-onset of PTSD due to combat exposure mostly found in former military deployed personnel [13, 14]. The impact of deployment is not only short term due to absence of the military personnel from home but is long-term [15].

When a parent with PTSD attempts to cope by withdrawing from social contact, they can present with emotional numbing, becoming irritable, hostile and show a lack empathy for their child [16–18]. As section 2.4.2 of the DSM-5 manual for PTSD notes, irritability and anxiety are two factors for the presentation of PTSD. It is these presentations which limit a person's problem-solving skills and conflict management [19], alongside poor spousal connectedness regarding parenting [20]. Problem solving, conflict management and connectedness for parenting are the foundations for promoting positive outcomes for the emotional growth of children [21].

Over the last decade, children and young people from Danish military families, have also experienced the constant negotiation of transition throughout their military life, in coping with cumulative stress and anxieties during parental separation in the context of war time danger [15]. Due to the ferocity and frequency of deployment many children and young people have experienced this type of separation multiple times, from birth to adolescence [22–25]. Multiple separations have been found to impact the psychological well-being of military children and young people, with reports of a decrease in children's resilience, an increase in poorer at home parental mental health, depression and internalising problems [22, 26–31].

A systematic review carried out by Søndergaard et al. [32] regarding the mobility and transition of military personnel and their families concluded that there is little research exploring military dependent children and young people's experiences of mobility and transition. This point is later supported by Cramm et al. [33] who also noted there is a need to understand how the transition from the military effects the health and well-being of the transitioning family. Consequently, the aim of this paper is to capture the narratives of children and young people from military families, to understand the experience of growing up in a military family and the impact of parental PTSD.

## Method

### Ethics statement

Ethical approval was obtained through Northumbria Universities ethics committee approval number 17425. The recruitment of young people within this study commenced on 1st August

2019 and was completed on the 6<sup>th</sup> August 2019. Due to the participants being under 18 written consent was gained from their parent/guardian, and as best practice written, and continual verbal consent was also gained from the young person who participated.

This study was carried out as part of a larger project aiming to understand the experiences of children and young people from military families and to critically evaluate the impact of a five-day residential camp as a psychosocial intervention on the emotional well-being, decision-making, confidence, resilience and self-esteem of military children and young people (see Authors, 2020 for full report) [34].

Acknowledging the researcher's interpretivist philosophical stance, the study seeks to understand participants' subjective experiences within their social context. As a reflexive process, the research team recognises the potential influence of personal biases, the role of language translation, and the impact of social desirability bias in shaping the data. The study is grounded in the belief that meaning is co-constructed through participant-researcher interactions, with a critical evaluation of the camp's effect on emotional well-being, resilience, and self-esteem, while remaining mindful of the limitations inherent in qualitative research.

Due to the participants being under 18 written consent was gained from their parent, and as best practice written, and continual verbal consent was also gained from the young person who participated.

Ten young people were recruited from the five-day residential camp in Denmark ran by Støt Soldater & Pårørende (SSOP) a Danish military charity for children of military veterans. Participants were aged between 12 and 19 (mean = 15.00, SD = 2.54), six were female and four were male. All participants had at least one parent who was a veteran with a self-reported diagnosis of PTSD at the time of the camp. A Danish military veteran is "a person who has been deployed in international operations at least once, on the grounds of a decision made by Folketinget, the Danish Government or a minister" [35 p.1]. Thus, participants' parent(s) were a mixture of those still serving in the Danish military and those who had left.

Semi-structured interviews were carried out to explore and understand young people's experiences of growing up with a veteran parent with PTSD. Interviews lasted between around 30 minutes and took place with an interpreter to translate the questions and the narratives of the participants taking part (if required). All interviews were recorded and transcribed to ensure robust data capture. The transcriptions were entered into NVivo and analysed using Braun and Clarke [36] six steps of thematic analysis: familiarisation, coding, generating themes, reviewing themes, defining and naming themes and writing up.

## Results

The young people within the study discussed what being a child from a Danish military family was like. These narratives shaped the context of the 'stories shared and told' [37]. Analysis of the participants narratives allowed for the construction of collective themes. These themes included military life and deployment, parental mental health, young people's emotional health and well-being and connectedness.

### Military life and deployment

Participants discussed in detail what life was like growing up in a military family, and how their life compared to their civilian friends. Participants also described how they felt parental deployment had affected their emotional health and well-being, their connection with family and friends, whilst also exploring the change in relationship with their veteran parent. Despite differences to their peers' lives, this was normalised, although many acknowledged that certain experiences made growing up in a military family difficult.

*"It's like a normal family, buts it's still very hard."*

*(p14, female, aged 12–14)*

*"Well, for me, it's been like that is how I grew up so for me there has been no difference, I don't know how it would be otherwise. So, for me it's like that's how I grew up."*

*(p48, female, aged 15–19)*

Whilst many of the participants explained growing up within a military family was normal, a leading thread throughout all the narratives was the psychological and practical difficulties with parental deployment. Participants described a loss of connection to their serving parent and the implications this loss had on their activities of daily living as a result.

*"I miss him a lot. Something is missing at home, we have a big house, and when someone is missing it's like empty."*

*(p25, female, aged 12–14)*

*"It's difficult I would say because people don't really get how it is when your dad is going out [Conflict], they think oh he is coming back, but you never know, that's scary not to know if he is going to come back or not."*

*(p43, female, aged 15–19)*

The experience of deployment was different for each participant, dependant on the length and frequency of deployment. This presented multiple complications within the home environment and dynamics as a result of the health and well-being of the 'stay at-home' parent and the integration of the returning parent. Many participants could see a change in their parent due to the symptomology of PTSD or other adjustment issues, with many participants remembering how their parent was prior to deployment.

*"When I was younger at school he would run in and say hello to us and just have this wink in his eye and a happy disposition. He always smiled, but now he is like, he still comes to us, but that wink is not there. He is not smiling that much, and we have to kind of fight for it to get him to smile."*

*(p43, female, aged 15–19)*

In addition to recognising a change in their parent, the impact of this change on the dynamics of the wider family was highlighted, suggesting extra pressure was placed on other family members.

*"It's stressed, my mum is the only one who is actually going to work. My dad had a job, but he is medically discharged."*

*(p33, female, aged 12–14)*

The insight from these young people is supported by a growing evidence base looking at transitioning back into the family home post combat [33, 38, 39]. Moreover, these young people demonstrate that the effect of deployment can be long-term and not restricted to those still serving, but also those who have left the military. Therefore, it is important when supporting military families, to consider military families in terms of the short, medium, and long-term effects military life can have on a family, during and beyond service.

## Parental mental health

The effect of military life on the participants' family; especially regarding deployment to a combat zone and the resulting impact of the serving parent's mental health was discussed by participants. The focus of theses narratives was on how parental mental health affects the home environment and familial relationships.

Parental mental health and parent-child relationships varied between mother and father. Of interest, the participants appeared to have a better relationship with their mother in comparison to their father, regardless of whether or not the mother presented with PTSD symptoms.

> ". . .dad has PTSD, and I have grown up with that. It was long before I was born, he got that diagnosis. There is a lot of conflicts at home deciding who is going to do what. My dad needs a lot of space he usually can't get it as we have four in our family."

> (p33, female, aged 12–14)

> "He has PTSD—It's been hard at times. As I say it has its ups and downs and it's definitely harder to live with him when he has his downs but yeah its, its hard you know, you have to be careful with what you say and what you doing and all of that and if he is asleep or anything you have to be quiet in the house so there is a lot of things you have to think about when he is doing stuff."

> (p48, female, aged 15–19)

The above quotes indicate that many participants were living in homes which were potentially complex and anxiety provoking. Interestingly, participants discussed their mother's mental health in a more compassionate way, focussing on describing the actual presentation of their poor mental health.

> "I just think my mum is sick. She has I don't know how to say it in English, but PTSD. . .mum has anger issues it's like an attack, anger yeah."

> (p17, female, aged 12–14)

> "She is trying to keep it down and simple but sometimes it goes bat shit crazy."

> (p28, male, aged 12–14)

In the study, participants focused less on how their mother's poor mental health impacted themselves or the rest of their family, and instead provided more descriptive accounts of their home environment.

The participants, whose mother was a veteran, did not talk in as much detail about their parent's mental health compared to those whose father was a veteran. Of note, the narratives from the participants regarding their father and the relationship with the family, had a greater focus on military conflicts, and the anticipation of these. For example, 'walking on eggshells' was a term often used, with participants reporting trying to second-guess how their actions could affect the actions of their father, in order to avoid additional conflict.

Within the narratives, there was a high presentation of conflict within the family. Participants highlighted that conflict was having a detrimental impact on their emotional and mental health, their relationships with other family members, their friends, and the connection with their wider community.

*"Sometimes it's hard because we get angry at each other and sometimes we start to fight. . .you can't say it's easy because it's not easy being in a family where people have a symptom [PTSD]."*

*(p14, female, aged 12–14)*

*"It's making my whole family stressed. When my mum has to work and take care of all the things [Household] she never gets time alone and that affects us, me and my sister as she gets mad quickly. I have just grown up with a dad who can't do things like other dads can. Like play football in the yard, go to concerts, events. It's harder for him, and sometimes he just needs space to be—It's making my whole family stressed."*

*(p33, female, aged 12–14)*

All but one participant described the complexities of family life post-deployment and the effects parental mental health had on the family dynamic. Observations of the data highlighted high volumes of conflict in family relationships that went beyond the participants' description of their civilian counterparts.

## Young people's emotional health and well-being

The effects of deployment and the impact subsequent parental mental health had on participants' emotional health and well-being were considered, with participants highlighting anxiety, self-harming behaviours–such as hitting walls and misusing alcohol, and feelings of sadness. This theme around the emotional health and well-being of the participants demonstrated the long-term effects of military deployment and perceived loss of a 'secure' relationship with their veteran parent, beyond parental separation.

Most participants described living in a state of anticipation, forward thought planning and living in an environment not conducive to positive mental health.

*"Yes, if there was conflict the day before. I was anxious for getting up the next day [in case there was going to be more conflict]."*

*(p14, female, aged 12–14)*

*"It's hard. It's like being in a hurry, but a lot more. Mentally overthinking stuff. . .when I am in this situation, I know that I think a lot, I just don't know what, maybe about the situation, or what my dad has been through. . ."*

*(p33, female, aged 12–14)*

As described within multiple narratives and demonstrated above, it was more than living with the anticipation of conflict, participants had anxiety. The impact of worrying about a parent, with the participants always being on alert decreased their own abilities to function daily.

Many the participants explained feelings of sadness. There was, however, an apparent distinction between feeling sad because they missed a deployed parent and those whose sadness presented as low mood.

*"Sad—that's more like inside me, and no one can see it when I go to school every day, usually I am sad, and I keep up a façade and I don't show my emotions in school."*

*(p33, female, aged 12–14)*

These presentations of feelings of sadness had a wider impact. Specifically, it created emotional conflict for the participants due to a perceived façade they upheld. This façade allowed them to cope with the complexities of their day-to-day lives and interactions with others, to avoid creating additional concerns for their family and their wider network but created a disconnect.

*"It is usually me who my friends come to with issues and I don't feel like I have anyone who I can go to with my issues, they just don't understand it."*

*(p33, female, aged 12–14)*

Several participants described maladaptive coping strategies to regulate their emotions. These young people reported difficulties in escaping their challenging and emotional environments. To cope with difficult situations at home, they disclosed that they would use harming behaviours to regulate their emotions.

*"I can hit the wall, and just do stupid stuff; it's hard to talk about. I have banged my head into the wall, multiple times. I have over thought and when I do that, I don't get any sleep. I take it out on the wall because I don't want to take it out on others."*

*(p33, female, aged 12–14)*

*"I cry a lot and find other ways to get that pain away, I punch my door, and I have tried cutting too to get the pain away."*

*(p43, female, aged 15–19)*

For some participants, this self-harming behaviour occurred on more than one occasion as an intervention to reduce emotional and physical pain.

## Connectedness

Some participants highlighted a difficulty in connecting with their civilian peers due to feeling 'different'. Their perceived difference centred on the participants' experience of growing up in a military family and having a parent with a mental health problem as a result of active combat.

*"I feel sad, and feel bad, I feel like I am not a human, it's like feeling different compared to others—When I am with my friends and their families, it feels like I am a different human."*

*(p14, female, aged 12–14)*

"I generally don't feel good about myself–I put on a facade."

*(p25, female, aged 12–14)*

This difficulty to connect resulted in feelings of being lonely, leading to participants feeling they had to put on a façade in order to interact and communicate with others. This was due to a perceived lack of understanding from others. Participants felt their civilian counterparts did not understand their lives within a military family, nor did they feel their friends understood what it was like to have a parent with PTSD due to military combat.

*"I am willing to talk about it, it is like this in my family, and my friends say they understand but that annoys me, because they don't. . . their dad does not have PTSD, their families are*

*not that stressed they cry because they don't get sugar on their breakfast. My friends are not part of the military community. It is usually me who my friends come to with issues and I don't feel like I have anyone who I can go to with my issues, they just don't understand it."*

*(p33, female, aged 12–14)*

*"Amongst my friends. Not really. They don't really understand it because they don't have very much experience in it, so they are not very understanding about it, but I try to get them to understand but I don't talk about it very much."*

*(p46, male, aged 18–19)*

Due to a lack of understanding from their peers who had no military experience, participants felt unable to connect and share life experiences, in relation to the military and poor parental mental health, with their friendship groups. Consequently, it was reported that participants' feelings were not validated by their friends, thus creating feelings of loneliness.

*"I didn't know what was happening. So, I think I kind of shut down and just yeah just was on my own."*

*(p48, female, aged 15–19)*

*"I was very sad, and felt lonely, because I couldn't be together with other people, not physically, but emotionally. I feel loved by my parents but not my friends."*

*(p28, male, aged 12–14)*

Ultimately, participants entered a continuous cycle where not feeling connected created feelings of loneliness, which in turn stopped them from connecting with their peers. By not connecting, many participants were seen as different, exacerbating their difficulties in connecting. For many participants, this often resulted in being bullied.

Several participants experienced bullying and acknowledged that this was as a result of their parent being in the military and being singled out as different to their peers. These participants were a minority within their school due to being part of a military family.

*"Well, when my dad was deployed it was very, very tough. I've always been bullied at school, and it definitely got worse when my dad was deployed. It definitely got worse with the bullying and all of that when my dad was deployed because I was more vulnerable at that time. I had kids actually specifically bullying me with my dad being deployed I had a kid coming up to me saying 'oh, hey your dad is out in the field because he doesn't want to be with you'. So, it definitely was hard for me at that time."*

*(p48, female, aged 15–19)*

*"Two schools merged and a boy in my class, he was always angry. He would call me bad things—you are different to the others, and stupid. It made me feel sad, and I was crying all the time at school. It's still hard to go to school, I am still sad, but I am not cry. It is better and I am just living with it, so I am just trying."*

*(p14, female, aged 12–14)*

The participants acknowledged that bullying had a detrimental effect on their emotional health and well-being and their ability to positively connect with their civilian peers. This lead one participant to disclose the following:

*"I started to ignore it and would not give a reaction to the bullies. That ended up with me getting ignored completely. I think it is better being bullied than be ignored. It's better to have someone to acknowledge you, even if that is fighting you than being completely invisible, in their eyes."*

*(p28, male, aged 12–14)*

All the above quotes signify a perceived lack of understanding from those whom the participants sought a genuine and positive connection with. Ultimately, this left many feeling lonely and resulted in bullying due to being different.

The key themes within this study identify the emotional and psychological effects of military life on the families of service members, particularly focusing on the impact of combat deployment and the mental health issues faced by the serving parent PTSD. Participants highlighted how the parent's mental health struggles affected the home environment and familial relationships, with children experiencing anxiety, fear, and sadness. Many participants felt a sense of loss and insecurity in their relationship with the veteran parent, which went beyond parental separation. Participants also struggled to connect with civilian peers due to their unique experiences of growing up in a military family, leading to feelings of isolation. Family conflict further strained emotional well-being, and the lack of understanding from friends, who couldn't relate to the military experience or mental health challenges, resulted in loneliness. The narrative illustrates the long-lasting impact of military service on family dynamics and individual emotional health.

## Discussion

Semi-structured interviews were carried out with Danish military children and young people to understand the experience of growing up in a military family and the impact of parental PTSD. This resulted in four main themes focussing on military life and deployment, parental mental health, young people's emotional health and well-being and connectedness.

Participants expressed that they generally perceived it as quite normal to grow up in military family, but the deployments affected their sense of cohesion with the deployed parental separation. The process of military deployment, be that one or multiple deployments and the re-integration of the serving member disrupted the family dynamic, creating an unsettled environment [25, 40]. Research has indicated that adolescents experience increased sadness, hopelessness, suicidal ideation, depression, withdrawal and changes in sleep and eating during parental deployment [27, 41]. Family relationships and prior functioning of the family and the child impact on the child's ability to cope and demonstrate a risk of using maladaptive behaviours to cope with parental deployment and re-integration [42]. Specifically, in this study, several participants discussed maladaptive coping mechanisms including self-harm.

For the participants in this study, it was especially difficult having a parent with PTSD which they felt had a direct impact on conflict within their family. The symptoms of PTSD are often associated with avoidance and hyperarousal, which has the potential to place additional stressors of the family. It is these symptoms which often interfere with a parent's ability to provide the child with appropriate warmth, affection and loving interactions [43]. In addition, a parent with PTSD can often struggle with appropriate parenting behaviours due to their own symptomatic presentations, specifically when meeting the emotional health and well-being of their children [44]. There is an abundance of literature which demonstrates the potential emotional harm living with parental PTSD has in children and young people [33, 45–48]. Within the participants stories they describe their parent with PTSD being disconnected and struggling to bond; with symptoms of emotional numbing and avoidance, which in turn

contributed to the struggles of re-integration and family functioning [49]. The fore mentioned descriptions of family life highlight the parent 'being there, but not there'.

The notion of 'absent parents' is a common theme within the work of McCormack and Devine [50] who explored what was meant by 'absent parents' in veterans (adult) children. One participant reflecting "If we were having emotional problems or things, it was always mum–it was never dad". 'Never Dad' is a term the participants within this cohort continually related to when describing the need for an emotional supportive parent [51]. Interestingly, there was a disparity in the way participants in this study discussed their relationships with their mother and father—more empathetic and descriptive of behaviour of their mother, whereas they focused on negative experiences and military history regarding their father.

Regardless, there was a consensus that parental PTSD had a negative effect on their own emotional health and well-being, specifically regarding increased anxiety, feelings of sadness and self-harm. When children and young people's emotional and psychological needs are unmet, they too are at risk of developing mental health issues [52]. Specifically, children with parents suffering from psychological injuries caused by military deployment are considered as being at higher risk [16, 44, 53]. This is unsurprising as the relationships between parental and youth emotional and behavioural health has been reported as reciprocal in nature, where the emotions and behaviour of one family member affects the family as a whole [31, 54, 55].

The majority of the participants felt that they constantly were having to predict the emotions and behaviour of their parent in order to act appropriately, which does not promote healthy psychological well-being. Constant worry about their parent led to participants feeling as though they could not relax and were constantly on 'edge'. Poorer well-being of military children has been well documented in US literature, where those of formerly deployed parents with psychological problems have higher rates of psychological, and behavioural problems, as well as psychiatric diagnoses than military children of those without psychological problems [28, 56, 57]. However, there has been minimal comparable research in a European setting, particularly with Danish military children.

There was great difficulty in creating relationships with other children and young people because participants felt they were different because of their family circumstances. This resulted in feelings of loneliness, a distance from others and feeling as though their peers did not understand them–several participants also experienced bullying for having a military parent. Across the lifespan, loneliness is often a transient experience, with no long-term effects [58]. According to Evolutionary Theory of Loneliness, this is due in part to the feelings of loneliness motivating a reconnection with other people [59]. Despite the willingness of the participants to connect with other people, they experienced difficulty in achieving this. Attending the residential camp gave them an opportunity to connect, however beyond that, it is unclear what the long-term impact of difficulties connecting to others would have on the emotional health and well-being of the participants.

## Recommendations for policy and practice

To address these issues, several recommendations for policy and practice have been proposed. These include enhancing mental health support for military families, with a focus on veterans and their children, and providing training programmes to help families cope with the emotional challenges of deployment and PTSD. Establishing peer support networks for military children, offering family-centred interventions, and integrating mental health programmes into schools and other community-based assets can also help mitigate the emotional impact. Additionally, conducting long-term research to better understand the needs of military connected children and young people, particularly in European contexts, and creating co-

produced reintegration programmes for veterans with PTSD and their families, will improve overall support for these families. Raising awareness among educators, healthcare providers, and the broader community about the unique challenges faced by military children is also crucial to ensuring positive emotional well-being form military connected children and young people.

## Limitations

There is paucity of research that has explored connectedness or a sense of belonging in military children and young people. It is unclear as to whether this is a new finding from this study or if this is as a result of participants being recruited from a residential camp with a primary focus on providing a safe place for military children and young people to connect to others with similar home lives. This study was conducted in a controlled environment (residential camp) and the participants were therefore a purposeful sample, it may not be possible to generalise the stories to populations outside of this environment or the military community whose parents do not have PTSD. The study provides valuable insights into the emotional and psychological challenges faced by military children, particularly those with parents suffering from PTSD. However, there are several directions for future research to deepen our understanding of these challenges. Longitudinal studies could explore the long-term effects of growing up in a military family affected by PTSD, examining how these experiences shape children's mental health as they transition into adulthood. Comparative studies across different military populations in various countries could help identify universal themes and context-specific factors influencing children's well-being. Additionally, exploring gender differences, the role of siblings and extended family, and the impact of peer support programs could provide a more comprehensive understanding of how military children cope with parental PTSD.

## Conclusion

PTSD has been described as a debilitating disorder which affects the person with PTSD and those who are close to them. As demonstrated throughout the mentioned literature and within the findings of this cohort, the symptoms of PTSD and its presentations such as emotional withdrawal, explosive anger, being on edge and aggressive outburst, have been seen to reduce the functioning of those suffering from PTSD. As demonstrated within the findings of this study, the presentation of PTSD from a veteran parent also has a significant effect on the family dynamic, its functioning which are described by the children within this study as living in fear of PTSD symptoms, not being able to plan outings or have friends stay due to not knowing what the PTSD presentation will be like on any given day. These responsive behaviours created sadness within this cohort of children who at times isolated themselves as they do not want to get in the way of the presentation of parental PTSD. It can be concluded, the presentation within the home of these military families is affecting the child's emotional health and well-being. The behavioural, social, and psychological effects on the young people within this study demonstrate the need for further intervention to support the ripple effects of combat related PTSD. This was highlighted by both King and Smith [60] and Watson and Osborne [34]. Within the fore mentioned literature, there is a need for prevention, early intervention and continued support for military families effected by combat related PTSD, and specifically the effects living within an environment of toxicity has on the child emotional health and well-being.

The findings of this study identify the critical need for identification, connection and early intervention, and continued support for military families affected by combat related PTSD. The emotional and psychological challenges faced by children in these families—ranging from

anxiety and self-harm to difficulties forming relationships—highlight the importance of proactive measures to address these issues before they escalate. Providing early mental health support, including tailored programs for both parents and children, can help mitigate the negative impact of PTSD on family dynamics and children's well-being.

Additionally, ongoing support throughout a child's development, particularly in the form of peer networks, community and school-based interventions, alongside systemic family-cantered therapy, is essential to ensure that military children do not carry the emotional burdens of military life and parental trauma into adulthood. These findings reinforce the need for comprehensive, long-term strategies to support military families, ensuring that both parents and children have access to the resources they need to thrive emotionally and psychologically.

## Author Contributions

**Conceptualization:** Paul Watson.

**Data curation:** Paul Watson.

**Formal analysis:** Paul Watson, Alison K. Osborne.

**Funding acquisition:** Paul Watson.

**Investigation:** Paul Watson, Alison K. Osborne.

**Methodology:** Paul Watson, Alison K. Osborne.

**Project administration:** Paul Watson.

**Supervision:** Paul Watson.

**Validation:** Paul Watson.

**Writing – original draft:** Paul Watson, Alison K. Osborne.

**Writing – review & editing:** Paul Watson, Alison K. Osborne.

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
