## [Decision Letter · Decision Letter 0]

2 Sep 2024

PMEN-D-24-00189

Being a Military Child in Denmark: Young people’s experiences of living with a parent with PTSD

PLOS Mental Health

Dear Dr. Watson,

Thank you for submitting your manuscript to PLOS Mental Health. After careful consideration, we feel that it has merit but does not fully meet PLOS Mental Health’s publication criteria as it currently stands. Therefore, we invite you to submit a revised version of the manuscript that addresses the points raised during the review process.

We look forward to receiving your revised manuscript.

Kind regards,

Jinjin Lu, Ph.D.

Academic Editor

PLOS Mental Health

Journal Requirements:

Additional Editor Comments (if provided):

Reviewers' comments:

Reviewer's Responses to Questions

**Comments to the Author**

1. Does this manuscript meet PLOS Mental Health’s publication criteria? Is the manuscript technically sound, and do the data support the conclusions? The manuscript must describe methodologically and ethically rigorous research with conclusions that are appropriately drawn based on the data presented.

Reviewer #1: Yes

Reviewer #2: Yes

2. Has the statistical analysis been performed appropriately and rigorously?

Reviewer #1: N/A

Reviewer #2: N/A

3. Have the authors made all data underlying the findings in their manuscript fully available (please refer to the Data Availability Statement at the start of the manuscript PDF file)?

Reviewer #1: Yes

Reviewer #2: Yes

4. Is the manuscript presented in an intelligible fashion and written in standard English?

Reviewer #1: Yes

Reviewer #2: No

5. Review Comments to the Author

Reviewer #1: The paper provides a clear overview of the research area and is well-structured. The writing is clear and concise, giving the reader an insight into PTSD and how it affects families/children.

Comments

1. Cover Page

- Abstract

check - critically evaluate the impact ...during a five-day

- Data availability - cannot

2. page 1 - same for abstract

3. Page 2

- paragraph (para) line 2 - - stereotyped group - condsider (e.g. rebellion)

- para 2 - line 2 - This includes surviving serious injuries due to improved weaponry....alongside enhanced rehabilitation.

4. Page 4 - Method

What is SSOP - a local charity?

5. Page 9 - para 3 lines 2/ 3 - The impact of worrying ... forward forecasting - forecasying perhaps

... and always being .... the 2nd part of the sentence can be reworded to flow better.

4th quote - repitition from page 5 (bottom page)

6. Page 13 - para 2 - line 2 sense of cohesion ...could add (and separation?)

- para 2, line 5 Research - r

line 10, missing comma after study

para 3 - line 4 - parent's ability

7. Page 14 - para 1 - line 4 - reword sentence - Never Dad and is something ...

It's always good to add in line numbers in submissions and as per guidelines.

This was interesting to read.

Reviewer #2: The paper explores the impact of parental combat-related PTSD on children and young people in Danish military families. It particularly focuses on their emotional health, family dynamics, and social connectedness. The authors use mixed methods (qualitative interviews and thematic analysis) allowing for an in-depth understanding of the experiences of young people.

The topic of the paper is particularly relevant because of the recent war in Europe.

The paper is generally well written and I did not find any major issues in it. However, there are a number of minor issues, including grammatical errors and awkward phrasings throughout the paper that need to be addressed. For instance, in the abstract and introduction, there are instances of unclear sentence structures that could detract from the paper’s overall clarity.

Therefore, I recommend publication pending revision. Below is a list of a few examples of unclear sentences and a few recommendations that could benefit the overall strength of the paper:

Minor issues:

1. Language use

Abstract: “Moreover, this article,

aims to understand the lived experiences of children and young people from military families and to

critically evaluate the impact of during a five-day residential camp in Denmark ran by Støt Soldater &

Pårørende (SSOP) as a psychosocial intervention on the emotional well-being, decision-making,

confidence, resilience and self-esteem of military children and young people.”

Methodology: Due to the participants being under 18 written consent was gained

from their parent, and as best practice written, and continual verbal consent was also gained from the

young person who participated.

Results: He always smiled, but now he is like, he still

comes into us, but that wink is not there.

2. Also please modify the abstract to clarify that the present paper did not evaluate the impact of the residential camp on the interview subjects.

3. The paper could benefit from a clearly marked limitations section.

4. The paper could greatly benefit from an elaboration on the types and frequency of self-harm these youth have engaged in.

5. The authors mention “PTSD has been described as a debilitating disorder which affects the person with PTSD and those who are close to them.” The paper could benefit from providing examples to the debilitating effects of PTSD. E.g. what are the reasons the interviewees are in the center, or what are some of their experiences that could be described as debilitating, rather than to referring to literature. Also, some of these examples could be given in case of family members of the children.

6. PLOS authors have the option to publish the peer review history of their article (what does this mean?). If published, this will include your full peer review and any attached files.

**Do you want your identity to be public for this peer review?** For information about this choice, including consent withdrawal, please see our Privacy Policy.

Reviewer #1: No

Reviewer #2: No

---

## [Editor Report · Decision Letter 1]

19 Sep 2024

PMEN-D-24-00189R1

Being a Military Child in Denmark: Young people’s experiences of living with a parent with PTSD

PLOS Mental Health

Dear Dr. Watson,

Thank you for submitting your manuscript to PLOS Mental Health. After careful consideration, we feel that it has merit but does not fully meet PLOS Mental Health’s publication criteria as it currently stands. Therefore, we invite you to submit a revised version of the manuscript that addresses the points raised during the review process.

We look forward to receiving your revised manuscript.

Kind regards,

Jinjin Lu, Ph.D.

Academic Editor

PLOS Mental Health
---

## [Decision Letter · Decision Letter 2]

8 Nov 2024

PMEN-D-24-00189R2

Being a Military Child in Denmark: Young people’s experiences of living with a parent with PTSD

PLOS Mental Health

Dear Dr. Watson,

Thank you for submitting your manuscript to PLOS Mental Health. After careful consideration, we feel that it has merit but does not fully meet PLOS Mental Health’s publication criteria as it currently stands. Therefore, we invite you to submit a revised version of the manuscript that addresses the points raised during the review process.

I would like to thank you on behalf of the journal for your revisions and your patience in awaiting this latest review.  Please find below the Reviewer's comments, and I would invite you to re-submit a minor revision.

We look forward to receiving your revised manuscript.

Kind regards,

Sergio A. Silverio, MPsycholsci (Hons), MSc

Academic Editor

PLOS Mental Health

Journal Requirements:

Additional Editor Comments (if provided):

Dear Authors,

I invite you to submit a minor revision of your manuscript to the journal, paying particular attention to the Reviewer's comments (below).  This constitutes a minor revision.

With thanks in advance and with kindest regards,

Sergio A. Silverio

Reviewers' comments:

Reviewer's Responses to Questions

**Comments to the Author**

1. If the authors have adequately addressed your comments raised in a previous round of review and you feel that this manuscript is now acceptable for publication, you may indicate that here to bypass the “Comments to the Author” section, enter your conflict of interest statement in the “Confidential to Editor” section, and submit your "Accept" recommendation.

Reviewer #3: All comments have been addressed

2. Does this manuscript meet PLOS Mental Health’s publication criteria? Is the manuscript technically sound, and do the data support the conclusions? The manuscript must describe methodologically and ethically rigorous research with conclusions that are appropriately drawn based on the data presented.

Reviewer #3: Yes

3. Has the statistical analysis been performed appropriately and rigorously?

Reviewer #3: N/A

4. Have the authors made all data underlying the findings in their manuscript fully available (please refer to the Data Availability Statement at the start of the manuscript PDF file)?

Reviewer #3: Yes

5. Is the manuscript presented in an intelligible fashion and written in standard English?

Reviewer #3: Yes

6. Review Comments to the Author

Reviewer #3: Dear authors, thank you for your submission. I found your study extremely interesting and a valuable and insightful contribution to the field, particularly in its exploration of the experiences of children from military families. The study is well-structured and provides a good overview of a very critical topic.

In the following feedback, I’ve offered some general recommendations to enhance clarity and impact, alongside specific suggestions to further strengthen the manuscript. Overall, I appreciated the relevance of the literature included, the robustness of your methods, and the clarity of your results.

GENERAL:-The manuscript requires a very minor proofread for spelling and grammar errors, for example the sentence on page 7: “The focus of theses narratives was on how parental mental health affects the home environment and familial relationships” – theses should be these, but I feel this sentence needs re-wording for clarity.

ABSTRACT:- Really clear and interesting. There is a grammatical error in the phrase "the impact of during a five-day residential camp." Correction: Remove "of" to read "the impact during a five-day residential camp."

INTRODUCTION:- You effectively highlight the paucity of research on military-dependent children’s experiences. It sets a really strong foundation for the significance of the study. The progression from discussing general stressors faced by military families to the specific impact of parental PTSD provided a really coherent narrative and I believe you’ve demonstrated some solid engagement with existing literature. Please reword the following sentence for clarity and precision: "Due to recent conflicts military personnel, their families and veterans face different physical and psychological issues." Suggestion: "Recent conflicts have led military personnel, their families, and veterans to face a variety of physical and psychological challenges." Since the study focuses on Danish military families, could you provide more specific information about Denmark’s military engagements in terms of the unique aspects of Danish military culture, and any support systems in place for military families.

MATERIALS AND METHODS:- The methods are well described, however, please could you include a statement of reflexivity, bias and your philosophical underpinning.

RESULTS:- The final sentence in the first paragraph on page six, “Importantly, a number of the participants’ parents had left the military before they were born, demonstrating the potential long-term effects of military deployment to a combat zone,” seems redundant amongst the quote that follows. However, the first sentence in the paragraph that follows “…many participants could see a change in their parent due to the symptomology of PTSD or other adjustment issues, with many participants remembering how their parent was prior to deployment” seems better placed in this section instead.

Could the following sentence on the bottom of p7: “Within the study, participants focussed less on the wider impact their mother’s poor mental health had on themselves, or the rest of their family, providing a more descriptive reality to their home environment” be re-worded to: "In the study, participants focused less on how their mother's poor mental health impacted themselves or the rest of their family, and instead provided more descriptive accounts of their home environment” for better clarity.

It appears that some themes have subthemes while others do not. To maintain consistency, it may be clearer to remove the subthemes and retain only the overarching themes. Additionally, the term 'conflict' is used both to refer to a specific foreign conflict (in brackets) and as a subtheme to describe 'conflict' within families. Based on the quotes, this seems to refer more to emotional conflict within the family, so clarifying this distinction would be helpful. However, as mentioned in my earlier comment, removing subthemes altogether may provide a more streamlined structure and avoid any inconsistencies.

Please ensure that there is no more than two quotes between each descriptive section.

The end of the results section would benefit from a central organising concept that unifies the key themes – military life and deployment, parental mental health, young people’s emotional health and wellbeing and connectedness – into an overarching narrative. It would help to illustrate the interconnected challenges faced by children from military families and emphasise the broader implications for support and intervention.

Please incorporate more quotes related to the theme of self-harm.

Remove any duplicate quotes and replace them, as the same quote should not be used across different themes. For example, under "Military Life and Deployment," the quote reads: “I miss him a lot. Something is missing at home, we have a big house, and when someone is missing it’s like empty.”

However, the same quote is partially repeated, with additional context, under "Young People’s Emotional Health and Well-Being":

“I miss him a lot. Something is missing at home, we have a big house, and when someone is missing it’s like empty. Kind of sad, something that is not there, which is normally there. It's tiresome, no matter how sad I am I know he won’t be home, so I try to not think about it, so I am not sad.”

This section could benefit from a quote to clarify its message:

These presentations of feelings of sadness had a wider impact. Specifically, it created emotional conflict for the participants due to a perceived façade they upheld. This façade allowed them to cope with the complexities of their day-to-day lives and interactions with others, to avoid creating additional concerns for their family and their wider network but created a disconnect.

DISCUSSION:-Your discussion is really comprehensive and engaging. The existing literature is both relevant and effectively highlights the importance of the study. I found the section on the impact of deployment on family cohesion and the challenges of re-integration particularly insightful and highly relevant to your paper. To strengthen this section further, consider adding a paragraph on the practical and policy implications of your findings, offering specific recommendations for interventions, support systems, or policy changes. For instance, you might suggest how schools or military support services could address issues such as loneliness and mental health among military-connected children.

Whilst limitations are acknowledged, it would be beneficial to include more detail on the study's strengths and future directions, particularly with respect to areas for further research (e.g., longitudinal studies, comparisons across military populations in different countries, etc.).

CONCLUSION:- The conclusion effectively summarises the key points, though it could be made more impactful by reiterating the significance of the study’s contributions. Emphasising the need for prevention, early intervention, and continued support for military families affected by PTSD would reinforce the importance of the findings.

7. PLOS authors have the option to publish the peer review history of their article (what does this mean?). If published, this will include your full peer review and any attached files.

**Do you want your identity to be public for this peer review?** For information about this choice, including consent withdrawal, please see our Privacy Policy.

Reviewer #3: No

---

## [Decision Letter · Decision Letter 3]

4 Dec 2024

Being a Military Child in Denmark: Young people’s experiences of living with a parent with PTSD

PMEN-D-24-00189R3

Dear Dr Watson,

We are pleased to inform you that your manuscript 'Being a Military Child in Denmark: Young people’s experiences of living with a parent with PTSD' has been provisionally accepted for publication in PLOS Mental Health.

Best regards,

Sergio A. Silverio, MPsycholsci (Hons), MSc

Academic Editor

PLOS Mental Health

Dear Authors,

It is with great pleasure that I write to accept your article. Just as our latest reviewer commented, this is important work, undertaken sensitively with a lot of thought - and provides the next steps for research and care for military children.

I would like to congratulate you on such a compelling article!

With very best wishes,

Sergio A. Silverio CPsychol

Reviewer Comments (if any, and for reference):

Reviewer's Responses to Questions

**Comments to the Author**

1. If the authors have adequately addressed your comments raised in a previous round of review and you feel that this manuscript is now acceptable for publication, you may indicate that here to bypass the “Comments to the Author” section, enter your conflict of interest statement in the “Confidential to Editor” section, and submit your "Accept" recommendation.

Reviewer #3: All comments have been addressed

2. Does this manuscript meet PLOS Mental Health’s publication criteria? Is the manuscript technically sound, and do the data support the conclusions? The manuscript must describe methodologically and ethically rigorous research with conclusions that are appropriately drawn based on the data presented.

Reviewer #3: Yes

3. Has the statistical analysis been performed appropriately and rigorously?

Reviewer #3: N/A

4. Have the authors made all data underlying the findings in their manuscript fully available (please refer to the Data Availability Statement at the start of the manuscript PDF file)?

Reviewer #3: Yes

5. Is the manuscript presented in an intelligible fashion and written in standard English?

Reviewer #3: Yes

6. Review Comments to the Author

Reviewer #3: Thank you for making the recommended changes to your manuscript. I thoroughly enjoyed being part of the review process and believe your publication makes a valuable contribution to important research on the lives of military children in Denmark and beyond. I found it particularly compelling, though undeniably poignant, to learn how these children navigate challenges such as anxiety, isolation, and even the risk of self-harm, whilst also dealing with the impact of parental PTSD on family dynamics. Your focus on the need for greater support and connection for these children and their families is crucial, and this study really highlights how much they could benefit from tailored help. Thank you for your hard work and for bringing attention to such a vital issue.

7. PLOS authors have the option to publish the peer review history of their article (what does this mean?). If published, this will include your full peer review and any attached files.

**Do you want your identity to be public for this peer review?** For information about this choice, including consent withdrawal, please see our Privacy Policy.

Reviewer #3: **Yes: **Elana Payne
